# Cognitive Deficits in Aging Related to Changes in Basal Forebrain Neuronal Activity

**DOI:** 10.3390/cells12111477

**Published:** 2023-05-25

**Authors:** Irene Chaves-Coira, Nuria García-Magro, Jonathan Zegarra-Valdivia, Ignacio Torres-Alemán, Ángel Núñez

**Affiliations:** 1Department of Anatomy, Histology and Neurosciences, Universidad Autónoma de Madrid, 28029 Madrid, Spain; irene.chaves@hotmail.es; 2Facultad de Ciencias de la Salud, Universidad Francisco de Vitoria, Pozuelo de Alarcón, 28223 Madrid, Spain; nuria.garcia@ufv.es; 3Achucarro Basque Center for Neuroscience, 48940 Leioa, Spain; jonathan.zegarra@achucarro.org (J.Z.-V.); ignacio.torres@achucarro.org (I.T.-A.); 4Facultad de Ciencias de la Salud, Universidad Señor de Sipán, Chiclayo 02001, Peru; 5Ikerbasque Science Foundation, 48009 Bilbao, Spain

**Keywords:** cholinergic neurons, electroencephalogram, IGF-I, sleep, neurodegenerative diseases

## Abstract

Aging is a physiological process accompanied by a decline in cognitive performance. The cholinergic neurons of the basal forebrain provide projections to the cortex that are directly engaged in many cognitive processes in mammals. In addition, basal forebrain neurons contribute to the generation of different rhythms in the EEG along the sleep/wakefulness cycle. The aim of this review is to provide an overview of recent advances grouped around the changes in basal forebrain activity during healthy aging. Elucidating the underlying mechanisms of brain function and their decline is especially relevant in today’s society as an increasingly aged population faces higher risks of developing neurodegenerative diseases such as Alzheimer’s disease. The profound age-related cognitive deficits and neurodegenerative diseases associated with basal forebrain dysfunction highlight the importance of investigating the aging of this brain region.

## 1. Introduction

A wide variety of age-dependent physiological and molecular changes have been described in the mammalian central nervous systems [1,2]. Healthy aging is associated with functional and structural changes in many brain regions, resulting in an important and selective decline of executive functions and attention, as well as working and episodic memories. This decline is associated with alterations in synaptic transmission, structural synaptic changes, and a loss of synaptic connections [3,4,5]. One of the most important changes in aging occurs in cortical activity, and one of the more important areas that control this activity is the cholinergic inputs from the basal forebrain (BF).

Existing evidence suggests that aging may lead to specific changes in cortical activity. For example, a reduction of synaptic input in the neocortex [6] and a loss of gray and white matter with aging has been described in humans [7,8,9], and a consistent loss of hippocampal synaptic connections has been also identified in old rodents [3,4]. In addition, numerous studies in both humans and laboratory animals suggest that wakefulness and sleep show changes with aging. The activity of the cholinergic neurons of the BF plays an important role in the control of the electroencephalogram (EEG) pattern that characterizes both awake state or slow wave and rapid eye movement (REM) sleep states. Thus, a reduction of cholinergic neuronal activity with aging will induce changes in the EEG. In this review, we aim to provide an overview of the changes in BF neuronal activity during healthy aging.

## 2. Electroencephalographic Changes Provoked by Aging

The EEG is an electrophysiological technique for recording electrical activity arising from the cortex. The EEG is thought to be primarily generated by cortical pyramidal neurons in the cerebral cortex that are oriented perpendicularly to the brain’s surface. The neural activity detectable by the EEG is the summation of the excitatory and inhibitory postsynaptic potentials of relatively large groups of neurons firing synchronously. EEG recordings may also be averaged, giving rise to evoked potentials and event-related potentials, which represent neural activity that is temporally related to a specific stimulus. They are used in clinical practice and research for the analysis of visual, auditory, somatosensory, and higher cognitive functioning. The importance of the EEG is that it detects the changes that occur in the cerebral cortex in different states due to the existence of characteristic waves in each state. For example, during sleep or anesthesia, waves from the delta frequency band (0.5–4 Hz) predominate; theta frequencies (4–8 Hz) mainly occur during drowsiness and REM sleep; finally, waves of alpha (8–12 Hz), beta (13–30 Hz), and gamma frequency (>30 Hz) bands predominate during wakefulness. These changes observed in the EEG represent population changes at the cortical level and are related to the changes that occur in the firing pattern of cortical and BF neurons [10,11].

In general, large-scale age-related structural changes such as cortical thinning, white matter degeneration, neurotransmitter dysregulation, and/or receptor distribution that occur in aging affect the EEG. Power reductions in the aging EEG have been described by numerous previous studies. In general, they described a reduction of the EEG power and a slow-down of the frequency bands, mainly in the alpha frequency band [12,13]. In addition, evoked potentials tend to increase latency and decrease amplitude with increasing age [13,14]. The EEG power reduction during aging could be explained by a number of factors. One interpretation would be that decreases in EEG power reflect underlying reductions in cortical current amplitudes due to reduced synaptic density, activity, synchronization, or some combination thereof [15]. On the other hand, significant cortical atrophy can occur during aging [16,17,18].

In addition, different pathways are of great importance in the control of cortical activity, such as cholinergic and GABAergic neurons from the BF or noradrenergic neurons from the locus coeruleus. In this review, we are going to focus on the BF and the changes that occur during aging that may explain the deficits in cortical information processing.

## 3. The Basal Forebrain

One of the most crucial structures controlling cortical activity is the BF. The BF provides most of the cholinergic projections to cortical and limbic structures [19,20]. Electrophysiological recordings in the BF, combined with EEG recordings, have indicated that cortical activation depends on BF inputs to the cortex [10,11,21,22]. Most of these effects have been explained by the release of acetylcholine (ACh) in the cortex during wakefulness or REM sleep [23,24,25,26].

The BF includes the medial septum, horizontal and vertical limbs of the diagonal band of Broca (HDB and VDB, respectively), the substantia innominata, and the nucleus basalis magnocellularis (nbM; Meynert basal magnocellular nucleus in humans), which provide most of the cholinergic innervation to the sensory, motor and prefrontal cortices, and hippocampus [27,28,29,30,31]. The BF also contains two other parallel projection systems to the cortex, one releasing GABA and the other glutamate [30,32,33,34]. Recently, the substantia innominate has been included in a large structure called the extended amygdala, which includes different structures involved in the control of behavior and emotions [35].

Cholinergic projection neurons of the BF are organized into overlapping groups of neurons that share common sets of projection targets. The first anatomical descriptions of the cholinergic projections were consistent with the notion of a diffuse pathway from the BF to the cortex that would explain why their activation caused a generalized increase in fast oscillations in the EEG [36,37,38]. However, new evidence concerning the BF system indicates the existence of a highly structured and topographic organization of efferent projections to sensory cortices [25,30,31,39,40,41,42]. The above-mentioned authors propose that cholinergic and noncholinergic projections to the neocortex are not diffuse but instead are organized into segregated or overlapping neuronal groups. For example, most of the neurons located in the HDB project to the primary somatosensory (S1) cortex AND maintain reciprocal projections with the prelimbic/infralimbic areas of the medial prefrontal cortex (Figure 1). However, the nbM has more widespread targets in the sensory-motor cortex and does not project to the prelimbic/infralimbic areas [28,29]. These findings pointed to the presence of specific neuronal networks between the BF and the cortex that may play different roles in the control of cortical activity. Consequently, the activation of small neuronal groups in BF facilitates responses in specific areas of the cortex and not in the entire structure.

The BF cholinergic projection neurons have extensive input to neocortex and hippocampus [37]. Cholinergic afferents are distributed at high density throughout all layers of the neocortex in rodents, with particularly high densities in cortical layers 1, 5, and 6 [43]. In the human neocortex, the highest density of cholinergic receptors is observed in superficial layers of most cortical areas [44,45,46]. Neurons within the medial septum or the diagonal band of Broca (MS/nDB) provide the major cholinergic innervation of the hippocampus [47].

Differences between neuronal groups of the BF are also observed in the existence of bilateral projections from the BF to the cortex [48]. The application of retrograde tracers in both hemispheres of the S1, auditory or visual, cortical areas showed labeled neurons in the ipsi- and contralateral areas of the HDB and substantia innominata. In contrast, the nucleus basalis magnocellularis only showed ipsilateral projections to the cortex. Accordingly, optogenetic stimulation of the HDB facilitated whisker responses in the S1 cortex of both hemispheres through activation of muscarinic cholinergic receptors. Thus, these findings have revealed that specific areas of the BF project bilaterally to sensory cortical areas, probably to contribute to the coordination of sensory processing in both hemispheres.

The BF cholinergic neurons participate in several cognitive processes that become impaired during aging or dementia, including Alzheimer’s disease [49,50]. Impairment of cortical activity during aging has been explained by reduced BF neuronal activity, mainly observed in the cholinergic system [49,51,52,53]. Studies in animals have shown that pharmacological inhibition or neurotoxic lesions of this region cause dramatic impairments of cortical activity, increasing EEG slow waves and reducing synaptic responses [54,55,56,57,58]. Thus, a decrease in cholinergic activity could be associated with age-related disorders in attention, memory storage, and retrieval [49,52,59]. Accordingly, different studies have demonstrated a decrease in the cholinergic cell number in the MS/nDB during aging that correlates with cognitive impairment [60,61]. In agreement with these data, treatments facilitating cholinergic transmission improve memory in impaired old animals [62,63,64].

Indeed, aging not only affects cholinergic neurons but also other neuronal types. Immunohistochemical characterization of the medial septum revealed a significant decrease in parvalbumin (PV)-positive cell bodies in aged animals [65]. In addition, the number of cells expressing GAD67 mRNA also decreased in these animals [61]. This decreased GABAergic inhibitory synaptic transmission in the BF has been associated with age-related cognitive impairment in rats [66] and could explain why cholinergic neurons showed higher spontaneous activity in older animals respect to younger ones (see above). However, the reduction of inhibitory synaptic transmission could be mitigated by an enhancement of GABAergic postsynaptic responses. The whole-cell current density of GABA-activated chloride currents was increased with age, consistent with an age-related increase in MS/nDB neuron response to GABA [67]. Therefore, the contradictory effects that are observed in old animals (increased spontaneous activity but decreased response to stimuli, or a decrease in the number of GABAergic neurons but an increase in the inhibitory synaptic response) could represent compensatory mechanisms to alleviate the cognitive deficits that occur in aging.

## 4. Neuronal Changes during Aging

Cholinergic signaling in the CNS provides important control over the dynamic of neuronal networks’ underlying information and cognitive processing. Even though BF neurons have been studied for many years, little work has addressed the changes of the electrophysiological properties of these cells during aging. An early report in MS/nDB neurons in vivo showed age-related differences in the firing pattern and axon conduction velocity [68], suggesting functional changes of intrinsic neuronal properties with age. However, whole-cell current clamp recordings from acutely dissociated neurons of BF neurons showed no apparent difference in the basic firing properties of young and aged neurons [66,69,70,71]. However, AMPA-induced current densities were significantly increased, whereas NMDA-induced currents were not affected during aging [72]. Therefore, although the basic electrophysiological properties of BF neurons that control their spike activity do not seem to change during aging, their synaptic responses are affected.

In addition, BF cholinergic neurons showed altered Ca^2+^ buffering that was associated with cognitive impairment [71,73]. MS/nDB neurons of aged rats had an increased current influx through voltage gated Ca^2+^ channels relative to those of young rats, suggesting a change in Ca^2+^ buffering in BF neurons during aging that may induce cell death. At the same time, these authors observed an increased rapid buffering capacity in aged neurons that may also represent a compensatory response to increased Ca^2+^ influx.

As is indicated above, the BF is one of the most crucial structures controlling cortical activity. Thus, changes at the cellular level of BF neurons during aging should provoke changes in the EEG recordings. In fact, power density across all frequency bands showed a general slowing of the EEG in older subjects. Elderly subjects also showed EEG oscillations 2- to 3-fold smaller in amplitude than younger adults during anesthesia [74]. It is well documented that alpha frequency (8–13 Hz) also changes with age. From early childhood up to puberty, alpha frequency increases but then starts to decline with age [75]. Theta rhythm recorded in the hippocampus arises from interactions between MS/nDB neurons and intra-hippocampal circuits [76]. Spectral analysis of EEG recordings also revealed that aging slows the theta rhythm [77]. Therefore, the changes observed in the EEG (slowing down of the oscillations and less amplitude) may reflect a decrease in cortical activity that would explain the decrease in the cognitive level of these people. This effect may be mainly due to a reduction of BF inputs.

In the hippocampus and neocortex, theta and gamma oscillations are the most prominent rhythms recorded in the awake state or during REM sleep [76,78,79,80,81,82]. Theta and gamma oscillations have been linked to hippocampal information processing and processes of learning and memory [83,84]. Both oscillations increase when the cholinergic projections from the BF are activated and interact (theta-gamma cross-frequency coupling) during the awake state or REM sleep [80,84].

In rodents, the interplay between theta and gamma oscillations plays a role in hippocampal information processing, which is reduced in old animals [85,86]. It has been indicated that theta-gamma coupling facilitates transfer of spatial information from the entorhinal cortex to CA1, facilitating many cognitive processes [82,87,88,89]. This hypothesis has been studied in animals and humans, and suggests that the diminution of the theta rhythm in older subjects [90] and decrements in the temporal precision in which gamma oscillation is coupled to a specific theta phase underlie the decline of associative memory in normal cognitive aging [91]. The age-related decline of theta-gamma coupling was reversed with physostigmine that is a reversible cholinesterase inhibitor, indicating that this process is facilitated by cholinergic inputs [85,86]. Taken together, these findings indicate that the reduction of BF neuronal activity with aging, mainly in the cholinergic system, induces an impairment in cortical activity that has consequences in many brain processes.

## 5. Insulin-like Growth Factor-I and the Aging Brain

IGF-I is considered a main component in the physiology of all tissues, including the brain [15,92,93,94,95,96,97]. It is well known that IGF-I plays a key role in learning and memory processes as a potent stimulator of neuronal activity. IGF-I increases the spontaneous firing rate, as well as the response to afferent stimulation in many target neurons [98,99,100,101,102,103], and modulates excitatory synaptic transmission in many areas of the brain [104,105,106,107,108,109]. IGF-I signaling modulates the activity of calcium-calmodulin-dependent kinase 2 alpha (CaMKIIα) and mitogen activated protein kinase (MAPK/ErK) through multiple signaling pathways [110,111,112]. These proteins (CaMKIIα and MAPK) regulate Ca^2+^ concentration and consequently, the modulation of synaptic plasticity such as long-term potentiation (LTP). In addition, it has also been shown that IGF-I modulates cortical inhibitory synaptic plasticity through activation of astrocytes [113].

Secretion of IGF-I declines over time until only low levels can be detected in individuals aged ≥60 years [114]. As IGF-I plays an important role in the regulation of cellular functions, a reduction in serum IGF-I levels in aging should lead to significant alterations in brain activity. The IGF-I receptor (IGF-IR) expression was dramatically decreased in pyramidal and granule cells of the hippocampus and in pyramidal cells of the somatosensory cortex in aged animals [115]. Thus, the activation of the IGF-IR/Akt/GSK3 intracellular pathway was reduced in old mice [116,117]. Consequently, a reduction of the IGF-I system may contribute to cognitive deficits, as has been suggested during healthy aging [1,98,99,114,116,118,119].

Findings from our laboratory showed that there was an important change in the BF activity during aging (Figure 2). We found that cholinergic neurons increased their spontaneous activity in old animals with respect to young animals. This finding was in agreement with a larger expression of c-fos in cholinergic neurons of old animals with respect to young animals [98]. However, the response to local injection of insulin-like growth factor-I (IGF-I) in the HDB nucleus decreased in old mice. This reduction in the IGF-I evoked effects in the BF, which had many consequences because IGF-I controls neuronal activity in this area, and increases neuronal activity and sensory processing in the cortex. An IGF-I injection in the BF elicited fast oscillatory activity in the electrocorticogram and facilitated whisker responses in the S1 cortex. These excitatory effects evoked by IGF-I decreased in old mice, probably due to a reduction in the number of IGF-I receptors, as was indicated by the immunohistochemistry studies [98,99].

According to this hypothesis, cholinergic-identified and non-identified neurons in the HDB nucleus showed a decreased response to IGF-I in old mice that provoked a reduction of cortical activity. Specifically, optogenetic stimulation of cholinergic neurons located in the HDB area facilitated whisker responses in the S1 cortex through activation of muscarinic receptors [30]; this facilitation of sensory cortical responses was decreased in old mice [98,99].

In addition, we have demonstrated that synaptic plasticity was reduced in old mice. A stimulation train of whiskers at 8 Hz induced a long-lasting response facilitation in layer 2/3 neurons of the S1 cortex recorded in young animals, but not in neurons recorded in old animals. Old mice also showed a reduction in the performance of a whisker discrimination task, when animals must discriminate between different textures in the arms of a Y-maze [99]. Local application of IGF-I in the S1 cortex improved the response to whisker stimulation and the long-lasting response facilitation in both young and old animals. These results suggest that the synaptic plasticity impairment observed in old animals may be due to a reduction of IGF-I inputs to cortical cells, as occurs in aging, but may be recovered if IGF-I levels are increased. In agreement with this suggestion, we have published that administration of IGF-I for 28 days through Alzet mini-pumps in old animals increased tactile evoked potentials in the S1 cortex [98]. Therefore, the results suggest that reduced levels of IGF-I in the BF and in the cortex contribute to reduced information processing in the cerebral cortex, helping to explain the cognitive deficits observed in aging.

## 6. Aging and Sleep

Previous animal studies have suggested that cholinergic BF neurons play an important role in sleep-wake regulation and are also implicated in cortical arousal [120,121,122]. Sleep disturbances are so common during aging and could be related to the impairment of BF activity. Healthy aging is associated with marked effects on sleep. In humans, aging has been associated with numerous and diverse changes in the EEG and sleep, such as increased sleep fragmentation, decreased total sleep time, sleep efficiency, and changes in the frequency bands of the EEG [123,124,125]. Numerous studies have also provided important insights into the global age-dependent alterations in sleep-wake and EEG architecture in rodents [126,127,128,129]. Aged mice have reduced wakefulness and did not sustain long periods of wake during the active phase. Decreased wake with aging was accompanied by an increase of non-REM sleep. However, there are notable discrepancies between species concerning the effects of aging. For example, slow-wave sleep is decreased in aged humans, whereas it is enhanced in aged mice [127,129].

In humans, spectral power analysis shows a reduction of the delta wave activity in the EEG (<4 Hz) in middle-aged and older compared to young adults [12,125,130]. These waves reflect a synchronized slow oscillation in cortico-thalamic networks, showing a depolarization and hyperpolarization oscillation of their membrane potential [131]. The reduction in slow wave amplitude correlates with neuroimaging studies showing age-related cortical thinning in frontal regions [17,132]. Non-pathological cortical thinning is thought to reflect cell body shrinkage and a reduction in the dendritic arborization and synaptic density of cortical neurons [9]. Rhythmic activity of BF neurons at delta frequency (<4 Hz) has been described, which correlates with cortical slow waves [14], suggesting that the activity of BF neurons not only contributes to the generation of fast EEG activities that are recorded in wakefulness but also to the slow oscillations that are recorded during anesthesia or the slow wave sleep. Thus, a reduction in the activity of BF neurons could produce a reduction of both fast and slow EEG activities, as has been described (see above).

## 7. Neurodegenerative Diseases in Aging

The prevalence of neurodegenerative diseases in the aging population is increasing due to the higher life expectancy. However, other factors may contribute the appearance of neurodegenerative diseases. Although aging is not the cause of many neurodegenerative diseases, it can aggravate them because of cellular senescence. Cellular senescence can occur at any life stage, from embryo to adulthood, although it is associated with the aging process [133].

A large body of evidence suggests that BF cholinergic neurons are selectively vulnerable to degeneration in Alzheimer’s disease (AD), mainly in the nucleus basalis of Meynert [134,135,136,137], as well as by a decline of cortical choline-acetyl transferase (ChAT) activity [135,138] and in aging (see below). These changes were also observed in mild cognitive impairment (MCI) patients [51], suggesting that neuronal damage in the BF induces cognitive deficits. Dysregulation of the cholinergic system is implicated in the cognitive decline associated with aging and dementia, including Alzheimer’s disease [49,51]. For example, an impaired cholinergic transmission has been associated with age-related disorders in attention and memory storage and retrieval [49,53,59,139].

Tau (t-tau) is a marker of axonal and neuronal damage [12,13] that increases with age [140,141,142]. Pathological tau is a hallmark of several neurodegenerative diseases, most notably Alzheimer’s disease. Recent findings suggest the view that plasma t-tau may serve to identify subclinical cerebral and cognitive deficits that occur in normal aging [142]. Misfolded tau deposition typically occurs first in the entorhinal cortex and hippocampus [143,144,145]. In addition, volume loss of the nucleus basalis of Meynert occurs in asymptomatic subjects with tauopathy and BF atrophy extends to the medial septum and vertical limb of the diagonal band Broca (Ch1–Ch2) [146].

Aβ accumulation in familial AD is mostly due to increased Aβ production by neurons, caused by mutations that potentiate the cleavage of amyloid precursor protein (APP) by β-secretase, or that alter the cleavage by γ-secretase to produce more Aβ42. Aβ42 is normally produced [147,148] and it can accumulate in the brain with aging because the clearance of Aβ is diminished with aging [149,150,151,152].

BF neurons are vulnerable to degeneration in the course of aging and in a number of other neurodegenerative conditions, such as Alzheimer’s disease, Parkinson’s disease, and Lewy body dementia [153,154]. Specific accumulation of amyloid-β has been observed in pyramidal neurons of the hippocampus, layer II of entorhinal cortex, and, notably, in BF neurons [149,155]. The reasons for the selective vulnerability of BF neurons in neurodegenerative disorders are unknown; however, alterations in intracellular Ca^2+^ homeostasis have been implicated in neuronal dysfunction (see above). Geula et al. [156] have described a decrease in expression of the Ca^2+^-binding protein calbindin-D28K in most of BF neurons during aging. BF neurons that lose calbindin are likely to increase phosphorylation of tau, increase accumulation of pre-tangles and neurofibrillary tangles, and cause their degeneration [157]. Therefore, intraneuronal amyloid-β accumulation in adult life and oligomerization during the aging process may contribute to the degeneration of BF cholinergic neurons in aging and in other neurodegenerative pathologies.

## 8. Conclusions and Future Directions

We have described in this review changes that occur in the BF during healthy aging which may be responsible for the decline in the cognitive level. Numerous and classical reports have shown that the cholinergic projections from the pons nuclei or from BF cholinergic neurons are fundamental in the control of cortical excitability and fundamental processes such as attention, learning, or memory. This type of neuron is very sensitive to neurodegeneration, such as that which occurs during healthy or pathological aging, causing progressive cognitive deterioration. Knowledge of the mechanisms by which the cholinergic system acts and its progressive deterioration with age can help to develop new therapies that reduce or delay this deterioration. For example, the diminution of the response to modulatory substances, such as IGF-I, may participate in the decreased neuronal activity observed in aging. A therapy that increases its levels in the brain would delay the cognitive deterioration produced by age.

## Figures and Tables

**Figure 1 cells-12-01477-f001:**
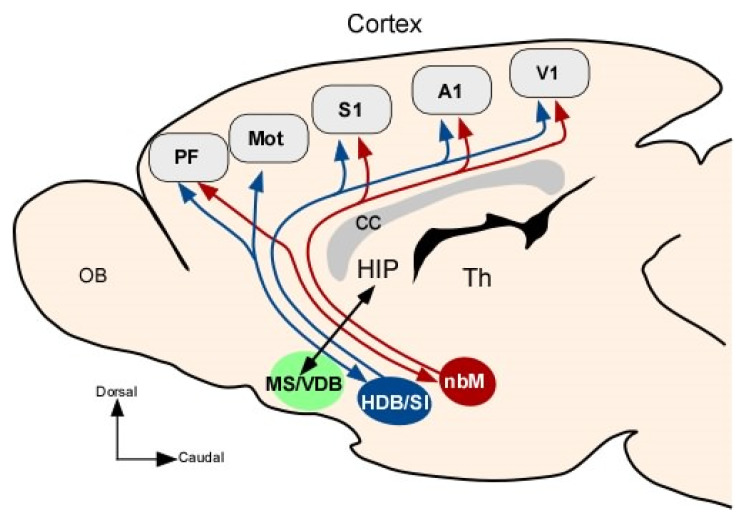
Diagram displaying the anatomical connections of the BF with the cortex. HDB/SI has preferential projections to sensory and prefrontal cortices, while the nbM nucleus projects to all sensory and motor areas in general and does not project to PF cortex. MS/VDB projects mainly to the hippocampal formation. Abbreviations: A1, primary auditory cortex; Mot, motor cortex; MS, medial septum; nbM, nucleus basalis magnocellularis; PF, prefrontal cortex; SI, substantia innominate; S1, primary somatosensory cortex; and V1, primary visual cortex.

**Figure 2 cells-12-01477-f002:**
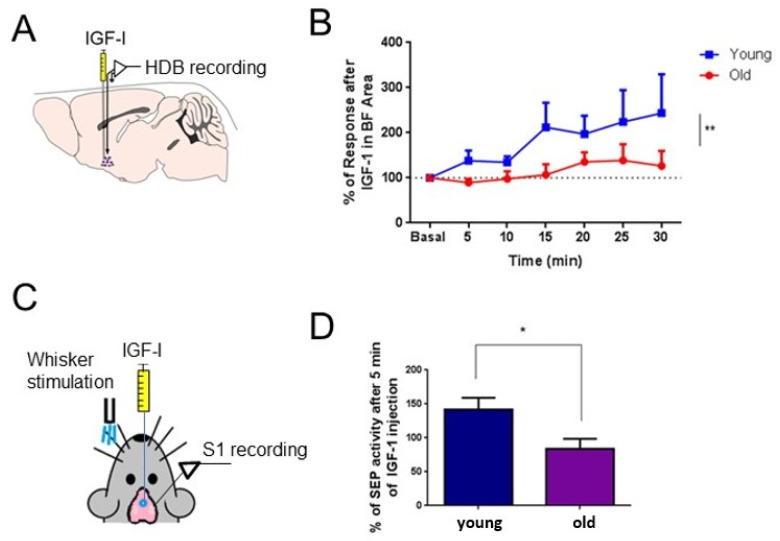
IGF-I facilitates neuronal activity in the BF and whisker responses in the S1 cortex. (**A**) A schematic diagram of the experimental design. A recording microelectrode was placed in the HDB nucleus; a cannula to inject IGF-I (10 nM; 0.2 μL) was also placed in the same area. (**B**) Plot of the HDB firing rate after IGF-I local injection. The firing rate is expressed as a percentage of basal responses at time 0 in both experimental groups. Young but not old mice responded to IGF-I in HDB nucleus (** *p* = 0.0044; Two-way ANOVA). (**C**) A schematic diagram of the experimental design. A recording microelectrode was placed in the S1 cortex and whiskers were stimulated with an air-puff (20 ms duration); a cannula to inject IGF-I was also placed in the HDB nucleus. (**D**) The area of the somatosensory evoked potential increase in young mice after local injection of IGF-I (10 nM; 0.2 μL) at 5 min after injection (141.9% ± 17.06 respect to control values), while the area was not affected in old mice (83.9% ± 14.71; * *p* = 0.0176; Unpaired *t*-test). Figure modified from Ref. [98].

## Data Availability

Our data will be provided by the corresponding author upon reasonable request.

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
