# Peer review of "Cognitive Deficits in Aging Related to Changes in Basal Forebrain Neuronal Activity"

_cells, 2023, doi:10.3390/cells12111477_

Round 1

Reviewer 1 Report

Manuscript #cells-2369158 ‘Cognitive Deficits in Aging Are Related with Changes in Basal Forebrain Neuronal Activity’ by I. Chaves-Coira et al., is an in-depth and insightful review surrounding aging and disturbances to the function of basal forebrain cholinergic neurons. The literature review is well annotated and extensive. The authors are commended for providing detail on the physiology associated basal forebrain cholinergic activity in aging paradigms. A few suggestions are raised to improve the submission. 

1. Title. The Reviewer recommends changing the title to ‘Cognitive Deficits in Aging Are Related to Changes in Basal Forebrain Neuronal Activity’.

2. EEG description section. Section 2 describing the anatomy and physiology of the basal forebrain is a noted strength of the submission. However, the authors do not go into any description of how EEGs and related recordings are performed i) in the basal forebrain or ii) in the cholinergic projection paths throughout the cortex, hippocampus, and subcortical structures in either awake humans or relevant animal models. Therefore, the Reviewer urges the authors to put in a new section 2 that gives an overview of the approaches the authors and the research community uses (e.g., EEG) to profile basal forebrain neurons and their projections.

3. Pathology in the basal forebrain and aging. The authors do not give any discourse on neuropathology (e.g., Abeta deposition/accumulation and tau pathology, among others) that is found within the aging brain, notably within the vulnerable basal forebrain and importantly, in the hippocampal and neocortical projection paths which undoubtedly interacts with cholinergic structure and function. The Reviewer suggests the authors add a section on this important topic. 

Minor point

1. A hard copyedit by a native English speaker is strongly recommended. 

A hard copyedit by a native English speaker is strongly recommended

Author Response

We like to thank you for the important and constructive suggestions that have greatly improved our manuscript.

Reviewer 1

  1. Title. The Reviewer recommends changing the title to ‘Cognitive Deficits in Aging Are Related to Changes in Basal Forebrain Neuronal Activity’.

We have changed the tittle of the manuscript as you suggested.

  1. EEG description section. Section 2 describing the anatomy and physiology of the basal forebrain is a noted strength of the submission. However, the authors do not go into any description of how EEGs and related recordings are performed i) in the basal forebrain or ii) in the cholinergic projection paths throughout the cortex, hippocampus, and subcortical structures in either awake humans or relevant animal models. Therefore, the Reviewer urges the authors to put in a new section 2 that gives an overview of the approaches the authors and the research community uses (e.g., EEG) to profile basal forebrain neurons and their projections.

We thank his constructive comments about the manuscript. We have included a new Section 2 about EEG description and basal forebrain neuronal firing (page 2).

  1. Pathology in the basal forebrain and aging. The authors do not give any discourse on neuropathology (e.g., Abeta deposition/accumulation and tau pathology, among others) that is found within the aging brain, notably within the vulnerable basal forebrain and importantly, in the hippocampal and neocortical projection paths which undoubtedly interacts with cholinergic structure and function. The Reviewer suggests the authors add a section on this important topic. 

We have included a new Section 7 about Pathology in the basal forebrain and aging as you suggested (page 9).

Minor point

  1. A hard copyedit by a native English speaker is strongly recommended. 

The manuscript has been revised by an English speaker. Thank you.

Reviewer 2 Report

It was my pleasure to review this article by Irene Chaves-Coira et al titled “Cognitive Deficits in Aging Are Related with Changes in Basal Forebrain Neuronal Activity”.

In this review, they summarized the recently accumulated knowledge about  the changes in basal forebrain activity during healthy aging. They highlighted the role of the basal forebrain cholinergic system in the mechanisms of brain function and its decline, reflecting on  its impact on our increasingly aging population facing higher risks of developing neurodegenerative diseases such as Alzheimer’s disease. The profound age-related cognitive deficits and neurodegenerative diseases associated with basal forebrain dysfunction highlight the importance of investigating the aging of this brain region. The article was well structured, contained a nice collection of up to date articles in the field on the rather complicated system, that is the cholinergic basal forebrain. I believe that this review provides a useful insight into where the research is standing at the moment, however I would have been interested to read the author’s opinion on where it should be heading, and what are the gaps left in the field to fill in, in order to have a better understanding about the cholinergic function during ageing. 

Minor comments: 

·      Fig 1. Details only the basalocortical projections and is missing the septohippocampal pathway, though that projection is also mentioned in the text. Might be worthwhile to include all the BF efferents?

·      In the most recent mouse brain atlas (Paxinos and Franklin's the Mouse Brain in Stereotaxic Coordinates 5th Edition - April 6, 2019), the substantia innominata is now called the extended amygdala. Or to be more precise, a part of the EA was formerly called the substantia innominata (SI). Please consider using the most up to date nomenclature. 

·      In vivo, in vitro should be always italicized  (line 169)

·      Ca2+ should be Ca2+ (line 206)

·      Any recommendation on the direction of future work in the area would be a nice addition into the conclusion. 

Author Response

We like to thank you for the important and constructive suggestions that have greatly improved our manuscript.

It was my pleasure to review this article by Irene Chaves-Coira et al titled “Cognitive Deficits in Aging Are Related with Changes in Basal Forebrain Neuronal Activity”.

We thank his helpful comments about the manuscript.

I believe that this review provides a useful insight into where the research is standing at the moment, however I would have been interested to read the author’s opinion on where it should be heading, and what are the gaps left in the field to fill in, in order to have a better understanding about the cholinergic function during ageing.

Thank you very much for your suggestion. We have included a new Section 8, “Conclusions and Future Directions” (page 10) with final considerations on the importance of the cholinergic system in many brain functions.

Minor comments: 

   Fig 1. Details only the basalocortical projections and is missing the septohippocampal pathway, though that projection is also mentioned in the text. Might be worthwhile to include all the BF efferents?

Figure 1 has been redone according to your suggestion. The septohipocampal pathway is now included.

   In the most recent mouse brain atlas (Paxinos and Franklin's the Mouse Brain in Stereotaxic Coordinates 5th Edition - April 6, 2019), the substantia innominata is now called the extended amygdala. Or to be more precise, a part of the EA was formerly called the substantia innominata (SI). Please consider using the most up to date nomenclature.

We think that many works use the term of “substantia innominate” so have kept this name in this manuscript. However, we have indicated in the text that SI is included in the extended amygdala area (page 3, lines 120-125).

  • In vivo, in vitro should be always italicized (line 169)

Thank you for noting the error (page 5 line 223).

      Ca2+ should be Ca2+ (line 206)

Corrected in all manuscript.

   Any recommendation on the direction of future work in the area would be a nice addition into the conclusion. 

As indicated above, a “Conclusions and Future Directions” section is included. Thank you for your suggestion.

Reviewer 3 Report

The manuscript by Chaves-Coira and colleagues aims at providing an overview of recent literature on age-driven changes in basal forebrain (BF) activity. The authors first discuss the anatomy and main projections of the BF. Then, they briefly highlight the importance of the BF in cognition and how reduced BF activity results in impaired cortical activity as well as in sleep alterations. Finally, they discuss the relevance of IGF-I to these processes.

The information provided is quite interesting and addresses a gap in knowledge. However, the manuscript would benefit of an entire re-writing. The information provided is very badly organized, creating a lot of confusion and masking the relevance of the topic. Below, I provide a few examples of specific areas to improve but the manuscript must be improved in other areas as well.

In the introduction, line 37, the authors should also cite the seminal work of Masliah (DOI: https://doi.org/10.1212/WNL.43.1_Part_1.192)

The narrative on the anatomy of the BF would benefit from adding some classical work.

In line 56, the authors could also cite the work of Gallagher and Colombo (doi: 10.1016/0959-4388(95)80022-0) as well as Mesulam (doi: 10.1016/0306-4522(83)90108-2)

Other citations could be included: http://dx.doi.org/10.1016/j.neuron.2016.09.006 ;

DOI: 10.1523/JNEUROSCI.14-03-01623.1994

    In figure 1, I suggest replacing B by NBM as it is the convention. Also, figure 1 is perplexing in that it challenges other representations of anatomical connections of cholinergic nuclei as presented for example in Lebois et al  2018 (https://doi.org/10.1016/j.neuropharm.2017.11.018) and in Paul et al 2015 (https://doi.org/10.3389/fnagi.2015.00043). The authors must consider better explaining or expanding their representation.

    The paragraphs on theta and gamma oscillations (lines 108-126) must be re-written and better put in context. These paragraphs are very convoluted and it is hard for the reader to follow the implications of this information in relation to BF changes in aging. Perhaps this topic should have its own section or be moved to the end of section 3.

    (https://doi.org/10.1038/srep05101). 

    Similarly, the paragraph on other cell types affected by aging (lines 151-165) seems out of place. Its link with the previous paragraph on IGF-I, signaled by “indeed”, suggests that a section is missing?

    Another example is the duplication of the information related to IGF-I in section 1 (lines 127-135) and section 4 (lines 197-236).

    The manuscript could benefit from additional grammatical editing once the sections have been re-organized.

    Author Response

    We like to thank you for the important and constructive suggestions that have greatly improved our manuscript.

    The information provided is quite interesting and addresses a gap in knowledge. However, the manuscript would benefit of an entire re-writing. The information provided is very badly organized, creating a lot of confusion and masking the relevance of the topic. Below, I provide a few examples of specific areas to improve but the manuscript must be improved in other areas as well.

    Thank you very much for your comments. We hope that the following changes improve our manuscript.

    In the introduction, line 37, the authors should also cite the seminal work of Masliah

    As suggested by the Reviewer, we now include this relevant work (page 2, line 47).

    The narrative on the anatomy of the BF would benefit from adding some classical work.

    We have included in this new version more classical reports (see ref. 19, 20, 28, 32 and 37).

    In line 56, the authors could also cite the work of Gallagher and Colombo (doi: 10.1016/0959-4388(95)80022-0) as well as Mesulam (doi: 10.1016/0306-4522(83)90108-2)

    Other citations could be included: http://dx.doi.org/10.1016/j.neuron.2016.09.006;

    DOI: 10.1523/JNEUROSCI.14-03-01623.1994

    We thank the reviewer for reminding us of these important references. We have included these relevant works in the new version of the manuscript by Gallagher and Colombo (page 4, lines 183-185), Mesulam (page 3, line 108), Ballinger (page 4, line 183) and Whalen (page 3, line 132).

    In figure 1, I suggest replacing B by NBM as it is the convention. Also, figure 1 is perplexing in that it challenges other representations of anatomical connections of cholinergic nuclei as presented for example in Lebois et al  2018 (https://doi.org/10.1016/j.neuropharm.2017.11.018) and in Paul et al 2015 (https://doi.org/10.3389/fnagi.2015.00043). The authors must consider better explaining or expanding their representation.

    We have redone Figure 1 according to your suggestion.

    The paragraphs on theta and gamma oscillations (lines 108-126) must be re-written and better put in context. These paragraphs are very convoluted and it is hard for the reader to follow the implications of this information in relation to BF changes in aging. Perhaps this topic should have its own section or be moved to the end of section 3.

    You are right. We move this paragraph  to section 4 and we have rewritten the paragraph (page 6, line 262-282). Thank you.

    (https://doi.org/10.1038/srep05101). 

    We think that this paper is out of the scope of this review.

    Similarly, the paragraph on other cell types affected by aging (lines 151-165) seems out of place. Its link with the previous paragraph on IGF-I, signaled by “indeed”, suggests that a section is missing?

    We have re-written this paragraph and it is included now in Section 5 (page 7, lines 319 and following)

    Another example is the duplication of the information related to IGF-I in section 1 (lines 127-135) and section 4 (lines 197-236).

    You are right, in this new version we have included the experiments with IGF-I in the same Section 5.

    Round 2

    Reviewer 3 Report

    The authors have implemented the suggested changes. Thank you.

    However, the addition of section 7 "Neurodegenerative Diseases in Aging", is perplexing. I can only suppose it was suggested by another reviewer. Still, this section needs major revision. The information provided is very superficial and not well-connected with the main subject. In particular while it is accepted that amyloid has a role in AD, how it influences basal forebrain neuronal activity is not clear from this section. Obviously, I am not asking for an extensive review of the subject as the body of literature related to this topic is quite vast. However, a few lines with key references must be added.

    I suggest a quick grammar check.

    Author Response

    However, the addition of section 7 "Neurodegenerative Diseases in Aging", is perplexing. I can only suppose it was suggested by another reviewer. Still, this section needs major revision. The information provided is very superficial and not well-connected with the main subject. In particular while it is accepted that amyloid has a role in AD, how it influences basal forebrain neuronal activity is not clear from this section. Obviously, I am not asking for an extensive review of the subject as the body of literature related to this topic is quite vast. However, a few lines with key references must be added.

    This section has been included at the suggestion of another reviewer.

    We have included in this new version amyloid-β accumulation in the basal forebrain and its effect in aging, as you suggest (page 10, second paragraph).

    We like to thank your suggestion.